# Approximate Size Targets Are Sufficient for Accurate Semantic Segmentation

## Abstract

We propose a new general form of image-level supervision for semantic segmentation based on approximate targets for the relative size of segments. At each training image, such targets are represented by a categorical distribution for the "expected" average prediction over the image pixels. We motivate the zero-avoiding variant of KL divergence as a general training loss for any segmentation architecture leading to quality on par with the full pixel-level supervision. However, our image-level supervision is significantly less expensive, it needs to know only an approximate fraction of an image occupied by each class. Such estimates are easy for a human annotator compared to pixel-accurate labeling. Our loss shows significant robustness to size target errors, which may even improve the generalization quality. The proposed size targets can be seen as an extension of the standard class tags, which correspond to non-zero size targets in each image. Using only a minimal amount of extra information, our supervision improves and simplifies the training. It works on standard segmentation architectures as is, unlike tag-based methods requiring complex specialized modifications and multi-stage training.

## 1   Introduction

Our image-level supervision approach applies to any semantic segmentation model and does not require any modification. It can be technically described in one paragraph, as follows. Soft-max prediction $S_p = (S_p^1, \ldots, S_p^K)$ at any pixel $p$ is a categorical distribution over $K$ classes, including background. At any image, the average prediction over all image pixels, denoted by set $\Omega$, is

$$\bar{S} := \frac{1}{|\Omega|} \sum_{p \in \Omega} S_p \tag{1}$$

where $\bar{S} = (\bar{S}^1, \ldots, \bar{S}^K)$ is also a categorical distribution over $K$ classes. It is an image-level prediction of the relative or normalized sizes (volume, area, or cardinality) of the objects in the image. We assume that training images have approximate size targets represented by categorical distributions $v = (v_k)_{k=1}^K$, e.g. $v = (0, .15, 0, \ldots, 0, .75)$ for the middle image in Fig. 1 if *"bird"* is the second class and *"background"* is the last. This representation also applies to multi-label images. For each training image, our *size-target loss*

$$L_{size} \;=\; KL(v \| \bar{S}) \;=\; \sum_k v_k \ln \frac{v_k}{\bar{S}^k} \tag{2}$$

is based on Kullback–Leibler ($KL$) divergence. Figure 2(b) shows some results for a generic segmentation network (ResNet101 [4] backbone) trained on PASCAL [5] using only image-level supervision with approximate size targets ($8\%$ mean relative errors). Our total loss is very simple: it combines size-target loss (2) and a common CRF loss (3) [6].

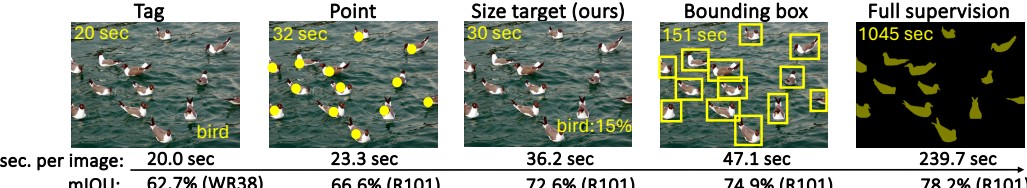

| | Tag | Point | Size target (ours) | Bounding box | Full supervision |
|---|---|---|---|---|---|
| sec. per image: | 20.0 sec | 23.3 sec | 36.2 sec | 47.1 sec | 239.7 sec |
| mIOU: | 62.7% (WR38) | 66.6% (R101) | 72.6% (R101) | 74.9% (R101) | 78.2% (R101) |

Figure 1: Supervision types for segmentation: labeling speed and accuracy on PASCAL. The top-left corner of each image shows its estimated labeling time based on observed instances. The table shows per-image labeling times averaged over the data and mean Intersection-over-Union (mIoU) for comparable end-to-end methods with similar ResNet backbones (ResNet101 or WideResNet38 [1]), for fairness. We obtained mIoU scores, except for the "tag" and "box" scores from [2] and [3]. Our supplemental materials detail evaluation of the labeling times and mIoU. For completeness, Tab.2 includes more complex architectures and multi-stage systems, e.g. for tags. This paper focuses on standard segmentation architectures for size supervision.

## 1.1 Overview of weakly-supervised segmentation

By *weakly-supervised* semantic segmentation we refer to all methods that do not use full pixel-precise ground truth (GT) masks for training. Such full supervision is overwhelmingly expensive for segmentation and is unrealistic for many practical purposes, see the right image in Fig. 1. There are many forms of weak supervision for semantic segmentation, e.g. based on partial pixel-level ground truth defined by "seeds" [6, 7], boxes [3], or image-level class-tags [2, 8, 9], see Fig. 1. It is also common to incorporate self-supervision based on various augmentation ideas and contrastive losses [10–12].

Lack of supervision also motivates unsupervised loss functions such as standard old-school regularization objectives for *low-level* segmentation or clustering. For example, many methods [13, 14, 12] use variants of K-means objective (squared errors) enforcing the compactness of each class representation. It is also very common to use CRF-based pairwise loss functions [6, 7] that encourage segment shape regularity and alignment to intensity contrast edges in each image [15]. The last point addresses the well-known limitation of standard segmentation networks that often output low-resolution segments. Intensity contrast edges on the high-resolution input image is a good low-level cue of an object boundary and it can improve the details and localization of the semantic segments.

Conditional or Markov random fields (CRF or MRF) are common basic examples of pairwise graphical models. The corresponding unsupervised loss functions can be formulated for continuous soft-max predictions $S_p$ produced by segmentation networks, e.g. [6, 7, 9]. Thus, it is natural to use relaxations of the standard discrete CRF/MRF models, such as *Potts* [16] or its *dense-CRF* version [17]. We use a bilinear relaxation of the general Potts model

$$L_{crf}(S) \quad = \quad \sum_k (\mathbf{1} - S^k)^\top W S^k \tag{3}$$

where $S := (S_p \,|\, p \in \Omega)$ is a field of all pixel-level soft-max predictions $S_p$ in a given image, and $S^k := (S_p^k \,|\, p \in \Omega)$ is a vector of all pixel predictions specifically for class $k$. Matrix $W = [w_{pq}]$ typically represents some given non-negative affinities $w_{pq}$ between pairs of pixels $p, q \in \Omega$. It is easy to interpret loss (3) assuming, for simplicity, that all pixels have confident *one-hot* predictions $S_p$ so that each $S^k$ is a binary indicator vector for segment $k$. The loss sums all weights $w_{pq}$ between the pixels in different segments. Thus, the weights are interpreted as discontinuity penalties. The loss minimizes the discontinuity costs [16].

In practice, affinity weights $w_{pq}$ are set close to 1 if two neighboring pixels $p, q$ have similar intensities, and weight $w_{pq}$ is set close to zero either when two pixels are far from each other on the pixel grid or if they have largely different intensities [6, 16, 17]. The affinity matrix $W$ could be arbitrarily dense or sparse, e.g. many zeros when representing a 4-connected pixel grid. The non-zero discontinuity costs between neighboring pixels are often set by a Gaussian kernel $w_{pq} = \exp \frac{-\|I_p - I_q\|^2}{2\sigma^2}$ of given bandwidth $\sigma$, which works as a soft threshold for detecting high-contrast intensity edges in the image. Thus, loss (3) encourages both the alignment of the segmentation boundary to contrast edges in the (high-resolution) input image and the shortness/regularity of this boundary.

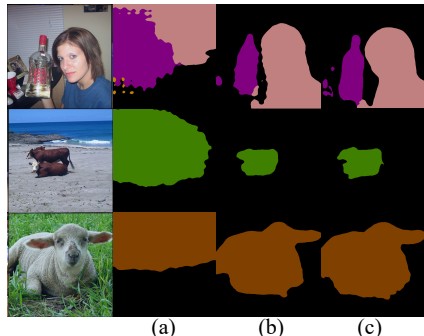

(a)          (b)          (c)

Figure 2: Semantic segmentation with standard DeepLabV3+(R101) segmentation models [18]: PASCAL validation results for training with (a) log-barrier (9) using class tags, (b) KL-divergence (2) using our approximate size targets, (c) cross-entropy with full (ground truth mask) supervision.

Weakly supervised segmentation methods may also use partial pixel-level ground truth where only some subset $Seeds \subset \Omega$ of image pixels has class labels [6, 7, 9]. In this case it is common to use *partial cross-entropy* loss

$$L_{pce}(S) \quad = \quad - \sum_{p \in Seeds} \ln S_p^{y_p} \tag{4}$$

where $y_p$ is the ground truth label at a seed pixel $p$.

## 1.2  Related balancing losses

Segmentation and classification methods often use "balancing" losses. In the context of classification, image-level predictions can be balanced over the whole training data. For segmentation problems, pixel-level predictions can be balanced within each training image. Our loss is an example of size balancing. Below we review some examples of related balancing loss functions used in prior work.

**Fully supervised classification.** It is common to modify the standard cross-entropy loss to account for unbalanced training data where some classes are represented more than others. One common example is *weighted cross-entropy*, e.g. defined in [19] for image-level predictions $S_i$ as

$$L_{wce}(S) \quad = \quad - \sum_{i \in D} w_{y_i} \ln S_i^{y_i} \tag{5}$$

where class weights $w_k \propto \frac{1}{1 - \beta^{v_k}}$ are motivated as a re-balancing factor based on the class distribution $v$ in the training dataset $D$ and $\beta$ is a hyper-parameter. In the fully supervised setting, the purpose of re-weighting cross-entropy is not to make the predictions even closer to the known labels, but to decrease over-fitting to over-represented classes, which improves the model's generality.

**Unsupervised classification.** In the context of clustering with soft-max models [20, 21] it is common to use *fairness* loss encouraging equal-size clusters. In this case, there is no ground truth and fairness is one of the discriminative properties enforced by the total loss in order to improve the model predictions on unlabeled training data. The fairness was motivated by information-theoretic arguments in [20] deriving it as a negative entropy of the data-set-level *average prediction* $\hat{S} := \frac{1}{|D|} \sum_{i \in D} S_i$ for dataset $D$

$$L_{fair}(\hat{S}) \quad = \quad -H(\hat{S}) \quad \equiv \quad \sum_k \hat{S}^k \ln \hat{S}^k$$

$$\stackrel{c}{=} \quad \sum_k \hat{S}^k \ln \frac{\hat{S}^k}{1/K} \quad \equiv \quad KL(\hat{S} \| u) \tag{6}$$

where $u = (\frac{1}{K}, \dots, \frac{1}{K})$ is a uniform categorical distribution, and symbol $\stackrel{c}{=}$ indicates that the equality is up to some additive constant independent of $\hat{S}$. Perona et al. [21] pointed out the equivalent KL-divergence formulation of the fairness in (6) and generalized it to a balanced partitioning constraint

$$L_{bal}(\hat{S}) \quad = \quad KL(\hat{S} \| v) \tag{7}$$

92   with any given prior distribution $v$ that could be different from uniform.

93   **Semantic segmentation with image-level supervision.** Most weakly-supervised semantic segmenta-
94   tion methods use losses based on segment sizes. This is particularly true for image-level supervision
95   techniques [2, 9, 22, 23]. Clearly, segments for tag classes should have positive sizes, and segments
96   for non-tag classes should have zero sizes.

97   Similarly to our paper, size-based constraints are often defined for the image-level *average prediction*
98   $\bar{S}$, see (1), computed from pixel-level predictions $S_p$. Many generalized forms of pixel-prediction
99   averaging can be found in the literature, where they are often referred to as *prediction pooling*. Some
100  decay parameter often provides a wide spectrum of options from basic averaging to max-pooling.
101  While the specific form of pooling matters, for simplicity, we discuss the corresponding balancing
102  loss functions assuming basic average prediction $\bar{S}$ in (1).

103  One of the earliest works on tag-supervised segmentation [9] uses *log-barriers* to "expand" tag
104  objects in each training image and to "suppress" the non-tag objects. Assuming image tags $T$, their
105  *suppression loss* is defined as

$$L_{suppress}(\bar{S}) \quad \propto \quad -\sum_{k \notin T} \ln(1 - \bar{S}^k) \tag{8}$$

106  encouraging each non-tag class to have zero average prediction $\bar{S}^k$, which implies zero predictions
107  $S_p^k$ at each pixel. Their *expansion loss*

$$L_{expand}(\bar{S}) \quad \propto \quad -\sum_{k \in T} \ln \bar{S}^k. \tag{9}$$

108  encourages positive average predictions $\bar{S}^k$ and non-trivial tag class segments.

109  We observe that the expansion loss (9) may have a bias to equal-size segments, as particularly evident
110  in the case of average predictions. Indeed, (9) implies

$$L_{expand}(\bar{S}) \quad \propto \quad KL(u_\mathrm{T} \| \bar{S}) \tag{10}$$

111  which is a special case of our size loss (2) when the size target $v = u_\mathrm{T}$ is a uniform distribution over
112  tag classes. The intention of the log barrier loss (9) is to push image-level size prediction $\bar{S}$ from
113  the boundaries of the probability simplex $\Delta_K$ corresponding to the zero-level for the tag classes
114  $T$. Figure 2(a) shows the results for training based on the total loss combining CRF loss (3) with
115  the log-barrier loss (9). Its unintended bias to equal-size segments (10) is obvious. Note that the
116  mentioned decay parameter used for generalized average predictions should reduce such bias.

117  Alternatively, it may be safer to use barriers for $\bar{S}$ like

$$L_{flat} \quad = \quad -\sum_{k \in T} \ln \max\{\bar{S}^k, \epsilon\} \tag{11}$$

118  that have flat bottoms to avoid unintended bias to some specific size target inside the probability
119  simplex $\Delta_K$. Similar thresholded barriers are common [22].

## 1.3   Contributions

121  In general, it would be great to have effective image-level supervision for segmentation that only uses
122  barriers like (9) or (11) since they do not require any specific size targets. This corresponds to tag-only
123  supervision. However, our empirical results for semantic segmentation using such barriers were
124  poor and comparable with those in [9]. A number of more recent semantic segmentation methods
125  for tag-level supervision have considerably improved such results [12, 24–30], but they introduce
126  significantly more complex multi-stage training procedures and various architectural modifications,
127  which makes such methods hard to replicate, generalize, or to understand the results. We are focused
128  on general easy-to-understand end-to-end training methods. Our main contributions are:

129  - We propose and evaluate a new general form of weak supervision, size targets. The size-
130    target supervision can be approximate and is relatively easy to get from human annotators.

131  - We propose the zero-avoiding variant of KL divergence as a general training loss, allowing
132    our end-to-end size-target approach to be integrated with any segmentation architecture.

133  - Comprehensive experiments with our size-target method demonstrate state-of-the-art perfor-
134    mance across multiple datasets using standard segmentation models typically employed for
135    full supervision, without any architectural modifications.

## 2 Size-target loss and its properties

Our proposed total loss is very simple

$$L_{total} := L_{size} + L_{crf} \tag{12}$$

where the two terms are our size-target loss (2) and standard CRF loss (3). The core new component is our size-target loss based on the *forward* KL-divergence. Our size-target loss (2) encourages specific target volumes for tag classes. Additionally, the size-target loss suppresses non-tag classes, encouraging zero volumes for classes not in the image. The CRF loss also contributes to the suppression of redundant classes. Therefore, unlike most prior work on image-level supervision for semantic segmentation, e.g. [9, 2, 12], we do not need separate suppression loss terms like (8). We validated this claim experimentally, they did not change the results.

Figure 3: *Forward* vs *reverse* KL divergence. Assuming binary classification $K = 2$, we can represent all possible probability distributions as points on the interval [0,1]. The solid curves illustrate our "strong" size constraint, i.e. the *forward* KL-divergence $KL(v\|\bar{S})$ for the average prediction $\bar{S}$. We show two examples of volumetric prior $v_1 = (0.9, 0.1)$ (blue curve) and $v_2 = (0.5, 0.5)$ (red curve). For comparison, the dashed curves represent reverse KL divergence $KL(\bar{S}\|v)$.

The size-target loss can also be integrated into other weakly-supervised settings, e.g. partial cross-entropy loss (4) commonly used for seeds. We show that using approximate size targets can significantly improve the seed-supervised segmentation in [6] when the seed lengths are short, see the right plot of Fig. 4.

$$L'_{total} := L_{size} + L_{crf} + L_{pce} \tag{13}$$

As is well known, KL divergence is asymmetric. In our work on image-level supervised segmentation, the order of the estimated and target distributions is crucial. The forward KL divergence possesses a zero-avoiding property, as illustrated in Fig. 3. Specifically, forward KL divergence imposes an infinite penalty when any class with a non-zero target is predicted as zero. In contrast, the penalty of the *reverse* KL divergence is finite and much weaker. When using reverse KL divergence, segmentation models tend to generate trivial solutions, predicting all pixels as the background class. This issue likely arises due to dataset imbalance, where the background class is prevalent. The zero-avoiding property of forward KL divergence ensures that segmentation models do not produce trivial solutions and predict all classes in the image tag sets.

## 3 Experiments

### 3.1 Experimental settings

**Datasets.** We evaluate our approach on three segmentation datasets: PASCAL VOC 2012 [5], MS COCO 2014 [31], and 2017 ACDC Challenge[1] [32]. The PASCAL dataset contains 21 classes. We adopt the augmented training set with 10,582 images [33], following the common practice [34, 9]. Validation and testing contain 1449 and 1456 images. Seed supervision of the PASCAL dataset is from [7]. COCO has 81 classes with 80K training and 40K validation images. ACDC Challenge is to segment the left ventricular endocardium. The training and validation sets contain 1674 and 228 images. The exact size targets are extracted from the ground truth masks.

**Approximate size targets.** We train segmentation models using approximate size targets $v = (v_k)_{k=1}^K$ generated for each image either by human annotators or by corrupting the exact size targets $\hat{v} = (\hat{v}_k)_{k=1}^K$ with different levels of noise. In all cases, we report the segmentation accuracy on validation data together with *mean relative error* (mRE) of the corresponding corrupted size targets. For each training image containing class $k$, the *relative error* for the size target $v_k$ is defined as

$$RE(v_k) = \frac{|v_k - \hat{v}_k|}{\hat{v}_k} \tag{14}$$

---

[1]https://www.creatis.insa-lyon.fr/Challenge/acdc/

where $\hat{v}_k$ is the exact size. mRE averages RE over all images and all classes. For human annotated size targets $v = (v_k)_{k=1}^{K}$, the relative size errors are computed directly from the definition (14).

When used, synthetic targets $v = (v_k)_{k=1}^{K}$ are generated by corrupting the exact targets $\hat{v} = (\hat{v}_k)_{k=1}^{K}$

$$v_k \longleftarrow (1 + \epsilon)\hat{v}_k \quad \text{for} \quad \epsilon \sim \mathcal{N}(0, \sigma) \tag{15}$$

where $\epsilon$ is white noise with standard deviation $\sigma$ controlling the level of corruption and operator $\longleftarrow$ represents re-normalization ensuring corrupted targets $(v_k)_{k=1}^{K}$ add up to one. Equation (15) defines random variable $v_k$ as a function of $\epsilon$. Thus, in this case, mRE can be analytically estimated from $\sigma$

$$mRE \; = \; E\left(\frac{|v_k - \hat{v}_k|}{\hat{v}_k}\right) \; \approx \; E(|\epsilon|) \; = \; \sqrt{\frac{2}{\pi}}\, \sigma \tag{16}$$

where $E$ is the expectation operator. The approximation in the middle uses (15) as an equality ignoring re-normalization of the corrupted sizes, and the last equality is a closed-form expression for the *mean absolute deviation* (MAD) of the Normal distribution $\mathcal{N}(0, \sigma)$.

**Evaluation metrics for segmentation.** We employ *mean Intersection-over-Union* (mIoU) as the evaluation criteria for PASCAL and COCO, and *mean Dice similarity coefficient* (DSC) for the ACDC dataset. The quality on the PASCAL test set is assessed on the online evaluation server.

**Implementation details.** We evaluate our approach with two types of ResNet-based [4] and one vision transformer (ViT) based [35] segmentation models on the PASCAL and COCO datasets. ResNet-based models follow the implementation of DeepLabV3+ [18] using the backbone of ResNet101 (R101) or the backbone of WideResNet-38 (WR38) [1]. For brevity, we name them R101-based or WR38-based DeepLabV3+ models. For the ViT-based network, We follow the implementation of Segmenter [36], adopting its ViT-B/16 backbone and linear decoder. For experiments on the ACDC datasets, we use MobileNetV2-based [37] DeepLabv3+ model. The R101, WR38, and MobileNetV2 backbones are ImageNet [38] pre-trained. ViT-B/16 backbone is pre-trained on ImageNet-21K [39] and fine-tuned on ImageNet-1k [38]. We directly evaluate our size-target approach on top of the standard architectures without any modification.

Images are resized to $512 \times 512$ for PASCAL and COCO, and $256 \times 256$ for ACDC. We employ color jittering and horizontal flipping for data augmentation. Segmentation models are trained with stochastic gradient descent on one RTX A6000 GPU with 48 GB GDDR6: 60 epochs for PASCAL and COCO, and 200 epochs for ACDC, with a polynomial learning rate scheduler (power of 0.9). Batch sizes are set to 16 for ResNet and 20 for ViT models on PASCAL, 12 on ACDC, and 12 (ResNet) and 16 (ViT) for MS COCO. The initial learning rate is 0.005 for ACDC and PASCAL's ResNet models, and 0.0005 for PASCAL's ViT models. The initial learning rate on COCO is 0.0005 for ResNet and 0.0001 for ViT models. Loss function (12) is employed for size-target supervision. Loss (13) is only used for seed supervision in Sec. 3.3. The implementation of CRF loss (3) is the same as [6]. We use $2\mathrm{e}^{-9}$ as the weight of the CRF term following the strategy in [6]. Size-target loss (2) and pCE (4) are used for medical images.

## 3.2 Robustness to Size Errors

We show the size targets can be approximate. The left plot in Fig. 4 illustrates the robustness of our approach to size errors. Segmentation models are trained with synthetic size targets subjected to varying levels of corruption, as defined in (15). The validation accuracy (solid red line) only drops slightly when $mRE$ (16) remains below 16%. The CRF loss (3) further enhances the robustness (solid blue line). When the relative error ($mRE$) is 4%, there is a noticeable increase in validation accuracy. The downward trend of the training accuracy (dashed blue line) suggests that the observed increases in validation accuracy at $mRE = 4\%$ stem from improved neural network generalization.

## 3.3 Enhancing seed-based segmentation with size targets

Our size-target approach can be integrated with partial ground truth mask supervision (seeds). The right plot in Fig. 4 demonstrates the results of seed-supervised semantic segmentation with and without size-target supervision. Size targets significantly enhance performance, especially when the seed lengths are short. Without size targets, segmentation performance degrades dramatically as the seed length decreases. Notably, when only one pixel is labeled for each object (seed length ratio = 0.0), size-target supervision boosts accuracy from 66.6% to 74%, approaching the performance of full seed supervision (seed length ratio = 1.0).

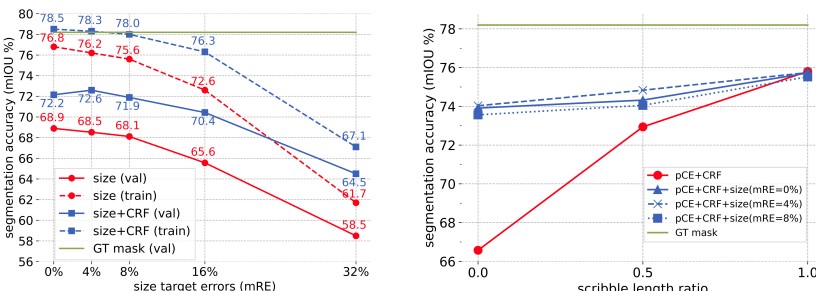

Figure 4: Segmentation results on the PASCAL dataset with R101-based DeeplabV3+ networks. The green bar in both plots indicates the segmentation accuracy for full ground truth masks (i.e. full supervision). The left plot shows the training and validation accuracy using approximate size targets. The segmentation is trained using losses (2) (red curve) or (12) (blue curve), where size targets are subject to various levels of corruption (15,16). The right plot shows validation accuracy for seed supervision of varying lengths with (blue curve) and without (red curve) using size targets. The line styles of the blue curves differentiate among various levels of corruption.

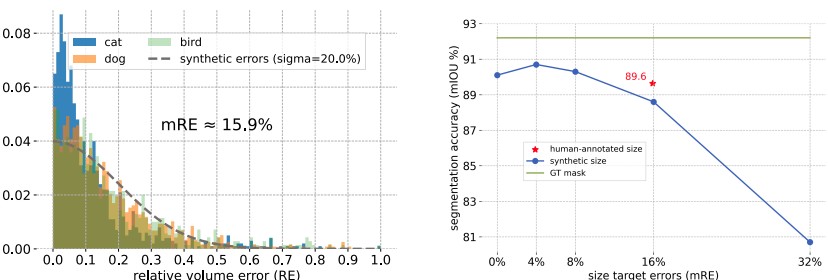

Figure 5: Left plot shows the quality of human annotations in terms of relative errors for the dog, cat, and bird classes within the PASCAl dataset. The histograms are normalized by the number of images in each class. The mean relative error for the three classes is 15.9%. For comparison, the dashed line shows the relative error distribution of synthetic size targets as defined in (15) for $\sigma = 20.0\%$ which aligns with the $mRE$ of 15.9%, see (16). The right plot presents 4-way multi-class (cat, dog, bird, and background) segmentation accuracy using human-annotated (red star at $mRE = 15.9\%$) and synthetic (blue curve) size targets, employing ResNet101-based DeeplabV3+ networks. Consistent with experiments in Sec. 3.2, synthetic size targets are generated at various levels of corruption. The green line indicates the segmentation accuracy of full supervision using ground truth masks.

## 3.4 Human-annotated size targets

**Annotation tool.** In this section, our approach is evaluated with size targets annotated by humans. We annotated training images for a subset of PASCAL classes, including cat, dog, and bird. A user interface with an assistance tool was developed to facilitate the annotation. The assistance tool overlays grid lines partitioning the image into $5 \times 4$ small rectangles or $3 \times 3$ large rectangles. Users can determine the size of a class in an image by counting rectangles (fractions allowed) or entering the percentage relative to the image size. Annotators can choose finer or coarser partitioning for each image depending on the object size. We evaluate relative errors with (14) for human annotations. Empirical evidence shows that annotators are approximately two times more accurate with the assistance tool, especially for small objects in the image. The last two columns of Table 1 report the annotation speed per image and mean relative error (14) for each class. The left plot in Fig. 5 shows the histograms of relative errors for human annotations. The histograms illustrate that annotated size errors are mostly below $10\%$, but occasional large mistakes (heavy tails) raise the mean error.

**Segmentation with human-annotated size.** Segmentation models trained with human-annotated size targets show robustness to human "heavy tail" errors. We compare the accuracy for human-annotated and synthetic size targets in the right plot of Fig. 5. The accuracy for human-annotated size (indicated by the red star in the plot) approaches 97.2% (89.6%/92.2%) of the full supervision performance, demonstrating that size-target approach is significantly robust to human errors. Binary segmentation accuracy for each class is reported in the shaded cells in Table 1. The performance of

| supervision | gt mask | gt size | human-annotated size | | |
|---|---|---|---|---|---|
| | mIoU | mIoU | mIoU | speed | mRE |
| cat | 90.6% | 88.8% | 88.0% | 12.6s | 12.3% |
| dog | 88.1% | 84.3% | 84.5% | 9.1s | 16.6% |
| bird | 88.8% | 86.2% | 86.4% | 15.2s | 20.1% |

Table 1: Human-annotated size targets. Two columns on the right show the average speed and relative error for each class we annotated. The shaded cells compare the accuracy of binary segmentation models trained with ground truth masks, ground truth size, and human-annotated size.

binary segmentation models trained with human-annotated size targets is comparable to those trained with precise size targets.

## 3.5 Comparison with the state-of-the-art methods

Our general training losses are applied to three standard architectures (R101-DeepLabV3+, WR38-DeepLabV3+, and ViT-Linear) for semantic segmentation as is, without any modification. Our results are highlighted in Table 2. The models are trained using synthetic size targets with an approximate mean relative error (mRE) of 8%. We chose this corruption level because its performance is close to human annotations, as shown in the right plot of Figure 5. Since our single-stage (end-to-end) approach is completely general, it is possible to use it in specialized architectures or complex training procedures. Likely, this would further improve the results, but this is not the focus of our work. The rest of Table 2 shows the results for semantic segmentation methods (of different complexities) for weak and full supervision. Methods are divided into multi-stage and single-stage methods, grouped by their backbones. Typical single-stage methods improve their results using complex architectural or training modifications such as additional training branches, extra refinement modules, or specialized training strategies. However, we achieve state-of-the-art using only standard segmentation architectures, commonly used in full supervision. The R101-based DeepLabV3+ model trained with approximate size targets approaches 92% (71.9/78.2) of its full supervision performance on PASCAL. The WR38-based DeepLabV3+ model trained with approximate size-target supervision surpasses other methods employing the same backbone by approximately 10%. Using the standard vision transformer architecture [36], the size-target approach achieves approximately 96% of the

| Backbone | Decoder | Architectural/training modification | Supervision | PASCAL | | COCO |
|---|---|---|---|---|---|---|
| | | | | Val | Test | Val |
| *Multi-stage methods* | | | | | | |
| R101 | DeepLabV3+ | MARS [40] arXiv'23 | tags | 77.7 | 77.2 | 49.4 |
| R101 | DeepLabV2 | MatLabel [41] ICCV'23 | tags | 73.0 | 72.7 | 45.6 |
| WR38 | LargeFOV | MCT [42] CVPR'22 | tags | 71.9 | 71.6 | 42.0 |
| WR38 | LargeFOV | MCTOCR [43] CVPR'23 | tags | 72.7 | 72.0 | 42.5 |
| SWIN | DeepLabV2 | ReCAM [44] CVPR'22 | tags | 71.8 | 72.2 | 47.9 |
| ViT-S | "Grad-clip" | WeakTr [26] arXiv'23 | tags | 78.4 | 79.0 | 50.3 |
| *Single-stage (end-to-end) methods* | | | | | | |
| R101 | DeeplabV3+ | - | size (8%) | 71.9 | 72.4 | 45.0 |
| R101 | DeeplabV3+ | - | full | 78.2 | 78.2 | 60.4 |
| WR38 | DeepLabV3+ | SSSS [2] CVPR'20 | tags | 62.7 | 64.3 | - |
| WR38 | Conv | RRM [45] AAAI'20 | tags | 62.6 | 62.9 | - |
| WR38 | DeeplabV3+ | - | size (8%) | 72.7 | 72.6 | - |
| ViT-B | LargeFOV | ToCo [28] CVPR'23 | tags | 71.1 | 72.2 | 42.3 |
| ViT-B | Conv | SeCo [29] arXiv'24 | tags | 74.0 | 73.8 | 46.7 |
| ViT-B | LargeFOV | CoSA [30] arXiv'24 | tags | 76.2 | 75.1 | 51.0 |
| ViT-B | Linear | - | size (8%) | 78.1 | 78.2 | 56.3 |
| ViT-B | Linear | - | full | 81.4 | 80.7 | - |

Table 2: Semantic segmentation results (mIoU%) on PASCAL and COCO. The supervision column indicates a form of supervision: image-level class *tags*, *size* targets (our highlighted results), or *full* supervision with pixel-accurate masks. The percentage after "size" is the accuracy (mRE) of our corrupted size targets (15,16). Our approach does not require any complex architectural modification or multi-stage training procedures needed for tag supervision, see "Modification" column.

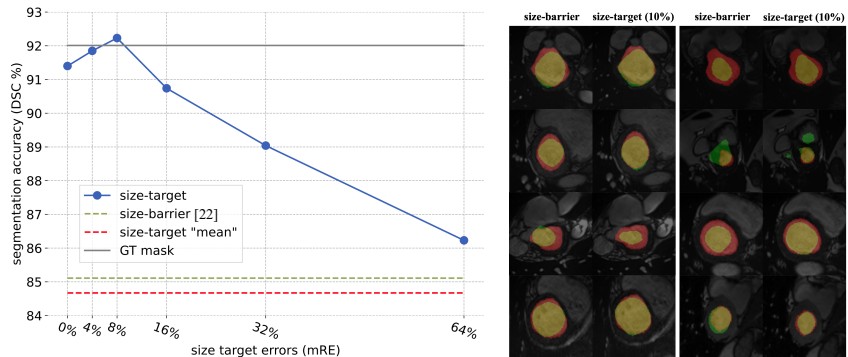

Figure 6: Size-targets (2) vs. size-barriers (17) on the ACDC dataset. The left plot shows the accuracy of the binary segmentation models (MobileNetV2-based DeeplabV3+) measured by DSC. The blue curve shows size-target accuracy with various levels of corruption. The dashed green line shows the accuracy of the size-barrier technique [22]. The dashed red line shows the accuracy using the mean size target for all training images. The gray line indicates the result of full supervision. The right image shows randomly selected qualitative results of size-barrier [22] and approximate size target ($mRE = 8\%$). Yellow shows true positive pixels, green is false positive, and red is false negatives.

full supervision performance on the Pascal dataset. Despite its simplicity, the size-target approach outperforms other complex single-stage methods on both datasets.

### 3.6 Medical data: size-target vs. size-barrier

Our method is also promising for medical image segmentation, benefiting from the consistency in object sizes across similar medical images, which healthcare professionals can easily estimate. We compare our size-target approach with the thresholded size-barrier technique [22], proposed for the weakly supervised medical image semantic segmentation. The size-barrier loss enforces inequality size constraints. Given the lower bound of each class, the thresholded size-barrier loss is

$$L_{flat\_sq}(S) \quad = \quad \sum_k \left( \max\{a_k - \bar{S}^k, 0\} \right)^2, \tag{17}$$

where $a_k$ is a lower bound of class $k$. We train binary segmentation models with a combination of partial cross-entropy loss (4) and size constraint loss: size-target (2) or size-barrier (17). Seeds used in the experiments are obtained using the same method provided in [22]. The object and background barrier, $a_{obj}$ and $a_{bg}$ are set based on [22]. In the size-barrier experiments, similarly to [22], we suppress the non-tag classes, using the loss $L_{sup}(S) = (\bar{S}^{obj})^2$. Conversely, size-target loss automatically suppresses non-tag classes as discussed in Sec. 2. The left plot in Fig. 6 displays the segmentation accuracy against different levels of size target corruption. Our size-target loss consistently outperforms size-barrier loss, maintaining its superiority even when using highly noisy size targets. The peak in the accuracy curve aligns with the experimental results in Sec. 3.2 and Sec. 3.4. The accuracy of the model trained using size targets with relative errors of 8% surpasses the full supervision performance. Additionally, using a fixed average size target across all training images can yield performance comparable to the size-barrier method, see the dashed red line in the left plot of Fig. 6. The right image in Fig. 6 shows qualitative examples of both methods.

## 4 Conclusions

We proposed a new image-level supervision for semantic segmentation: size targets. Such targets could be approximate. In fact, our results suggest that some errors can benefit generalization. The size annotation by humans requires little extra effort compared to the standard image-level tags and it is much cheaper than the full pixel-accurate ground truth masks. We proposed an effective size-target loss based on forward KL divergence between the soft size targets and the average prediction. In combination with the standard CRF-based regularization loss, our approximate size-target supervision on standard segmentation architectures (DeepLab and ViT) achieves state-of-the-art performance. Our general easy-to-understand approach outperforms significantly more complex weakly-supervised techniques based on model modifications and multi-stage training procedures.

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

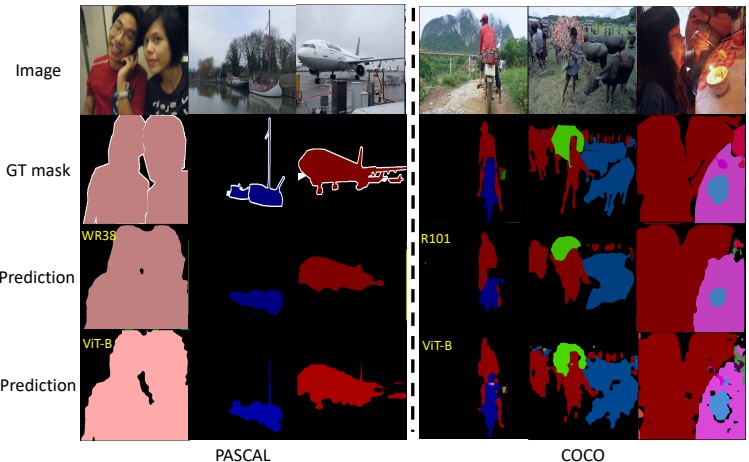

Figure 7: Segmentation examples using size-target supervision ($mRE = 8\%$). Model backbones are shown in the top-left corner of the predictions, see Table 2 for decoders.

## A    Appendix / supplemental material

### A.1    Labeling costs and accuracies reported in Figure 1

**Labelling costs.** Figure 1 in the paper shows labeling speed and accuracy for different forms of supervision on PASCAL VOC. The table at the bottom of Figure 1 shows ballpark estimates of average labeling time per image in the whole dataset. We use the data in [46], as well as Table 1 in the paper, and aggregate all labeling speeds from "per class", "per instance", or "per point" to "per image" using the average number of instances or classes in each image and the aggregation rules formulated in [46], see their Section 4. The top-left corner in each picture shows the corresponding estimated labeling times for the representative multi-instance image. All the labeling times are only rough estimates, but they are intuitive. The relative costs for point supervision seem underestimated, but they follow evaluation conventions detailed in [46].

**Accuracies.** The values of "point", "size target" and "full supervision" accuracy (mIOU%) are based on the experiments in the paper (Figure 4). We follow the learning rate scheme in DeepLabV3+ [18] for the training with full supervision. For fairness, we compare these with end-to-end methods using similar ResNet backbones in *tag*- [2] and *box*-supervision [3]. Typical SOTA methods for tag and box supervision use special architectural modifications, unlike our generic size-target loss, cannot be seamlessly plugged into any segmentation model.

### A.2    Qualitative results

Figure 7 presents the qualitative examples of our method on PASCAL (left) and COCO (right) validation sets. Despite size targets providing only image-level information, segmentation models can precisely identify object locations, eliminating the need for localization methods like CAM.

