# OpenReview forum: "Approximate Size Targets Are Sufficient for Accurate Semantic Segmentation"
_NeurIPS.cc/2024/Conference — Submitted to NeurIPS 2024_

### Official Review · Reviewer_thaL · 2024-07-02

**Soundness:** 3
**Presentation:** 3
**Contribution:** 3
**Rating:** 6
**Confidence:** 5

**Summary:**

This paper proposes a new weakly supervised semantic segmentation task. This task uses pixel-level categorical distribution as the label in the training stage. KL divergence is used as the training loss. Experiments on three public segmentation datasets show the effectiveness of the proposed method.

**Strengths:**

1.The proposed task is interesting. It provides the community another choice for segmentation with less annotation effort.

2.The proposed KL divergence loss is effective, demonstrated by experiments on three public datasets. It achieves performance comparable to methods using more expensive labels, like the box supervised one.

3.The proposed method is robust to size target error, which makes it more practical.

4.The writing is fluent and easy to follow.

**Weaknesses:**

1.Labeling effort on complex images. Images from PASCAL VOC (like Figure 1) are easy to annotate. It contains few classes and the background is generally clean. The density of target objects is low, and hence it’s also suitable for the proposed grid-based size target annotation way.

However, in practice, scenes are much more complex, with more classes, more crowded objects, and complex backgrounds. The authors are recommended to show the annotation effort on those images, like images from Cityscapes and ADE20K. I think when the scenes become more complex, the labeling effort will increase significantly. The labeling effort of size target will be much more than the tag way, since tagging will be less influenced in such cases.

2.Model performance on complex images. Similarly, it’s recommended to evaluate the model’s performance with the proposed loss on these complex datasets. This will give a more comprehensive understanding of the proposed method.

**Questions:**

Will the code, the labeling tool, and the labeled images be publicly available, so the community can use these tools to annotate their own datasets?

**Limitations:**

No negative societal impact.

---

> ### Author Rebuttal · Authors · 2024-08-05
>
> **Comment 1:** Labeling effort on complex images. Images from PASCAL VOC (like Figure 1) are easy to annotate. It contains few classes and the background is generally clean. The density of target objects is low, and hence it’s also suitable for the proposed grid-based size target annotation way.
>
> However, in practice, scenes are much more complex, with more classes, more crowded objects, and complex backgrounds. The authors are recommended to show the annotation effort on those images, like images from Cityscapes and ADE20K. I think when the scenes become more complex, the labeling effort will increase significantly. The labeling effort of size target will be much more than the tag way, since tagging will be less influenced in such cases. \
> **Response:** The image in Figure 1 is a selected example to showcase different forms of supervision and does not represent a typical image in PASCAL. The assumption that "images from PASCAL are easy to annotate" is questionable. PASCAL is highly diverse and includes images with complex backgrounds. While the number of categories in PASCAL is fewer than in datasets like Cityscapes and ADE20K, our size-annotation tool focuses the user on one category at a time, which helps to manage the complexity.
>
> We appreciate the suggestion to evaluate on more datasets. Given our limited resources, we chose to align with most prior works in WSSS that use PASCAL and COCO for evaluation. Additionally, we evaluated our method on a medical dataset to provide a comprehensive study. We acknowledge the value of assessing our method on more complex datasets like Cityscapes and ADE20K and will consider including these in future work.
>
> **Comment 2:** Model performance on complex images. Similarly, it’s recommended to evaluate the model’s performance with the proposed loss on these complex datasets. This will give a more comprehensive understanding of the proposed method. \
> **Response:** This comment is similar to the previous one. We appreciate the suggestion and will strive to accommodate it in future work.
>
> **Comment 3:** Will the code, the labeling tool, and the labeled images be publicly available, so the community can use these tools to annotate their datasets? \
> **Response:** Yes, the code, labeling tool, and labeled images will be posted.

---

> > ### Comment · Reviewer_thaL · 2024-08-13
> >
> > Thanks for the author's response. However, I think it's a consensus in the segmentation community that PASVAL VOC is much simpler than ADE20K and Cityscapes from various aspects, including the image resolution, the scene complexity, the annotation granularity. The segmentation performance for WSSS methods on these benchmarks also reveals this. Hence it is strongly recommended to experiment with these benchmarks, which can make this work more comprehensive.

---

> ### Author Response · Authors · 2024-08-13
> **on more complex datasets, etc...**
>
> Thanks for your feedback inspiring an interesting discussion. We would like to bounce back several related thoughts.
>
> **A [complex classes, not datasets]**: Your motivation for re-evaluating our ideas on more datasets is based on a speculation that human size annotation accuracy will decrease on more "complex datasets", whatever that may mean. Since we ask annotators to size only one class at a time, it makes sense to focus the discussion on some specific class that might be harder to size than classes on PASCAL. **Could you please name some particular class on Cityscapes or DE20K that you think is harder to size than “birds” and why you think so?** We do not see how image resolution or abstract “scene complexity” is relevant. If “annotation granularity” means high-frequency boundary details, they are mostly irrelevant for size estimation accuracy (due to “averaging”), unless many thin structures dominate the object shape, as often the case with birds.
>
> We agree that harder-to-size classes may exist, but it would be helpful to have a specific example from the datasets you suggested that would make this discussion less speculative. We believe that significantly higher human errors are possible for some **extreme** examples of classes that this discussion can identify (we can name them in limitations). This may also degrade the corresponding results. However, our results are sufficiently convincing for many representative objects that could be found in practical applications. Things work even for sufficiently challenging classes like birds. One should also keep in mind that **better assistance tools could be designed for specific complex classes**. In general, our results are meant to be a **proof-of-concept** and we believe they sufficiently demonstrate the potential of our novel ideas.
>
> Do you seriously doubt that our novel ideas could be useful?
>
> **B. [accuracy is an issue for all forms of supervision]**: For example, more complex objects lead to more mistakes for boxes, scribbles, and full pixel-masks in particualr. We saw many mistakes in the (so-called) ground truth masks in PASCAL (missing birds in a large group, ignored fine or thin shape details, e.g. in bikes, ambiguous categories like a "WV camper" car labeled as a bus, or toy labeled as a truck). That is, the accuracy of supervision is a general issue not limited to size annotations. Its full understanding is well beyond the scope of our work. in particular, likely problems with ground truth for complex classes (or datasets) may complicate the analysis of the effect of size annotation accuracy.
>
> **C. [room for future work]**: More labeled data and more experiments are always good, but we propose to leave some for future work (e.g. for a journal version), particularly because it would require a significant amount of annotation just to identify some specific “hard” classes.
>
> **E. [prior WSSS literature mostly use PASCAL. Maybe it is sufficient for concept proving?]** Note that the most significant/influential/conceptual prior work on WSSS are focused on PASCAL, to the best of our knowledge. While one can argue that other datasets are more complex (in some ways), it is more complex for all methods – full or weakly supervised.  In particular, are there examples in prior WSSS work where comparison on PASCAL is reversed on other datasets?

---

### Official Review · Reviewer_AEBs · 2024-07-12

**Soundness:** 2
**Presentation:** 2
**Contribution:** 2
**Rating:** 5
**Confidence:** 2

**Summary:**

The paper titled "Approximate Size Targets Are Sufficient for Accurate Semantic Segmentation" proposes a novel method of semantic segmentation that leverages approximate size targets instead of full pixel-level supervision. The method involves using categorical distributions to represent the expected average prediction over image pixels, utilizing the zero-avoiding variant of KL divergence as a training loss. The approach aims to reduce annotation costs while maintaining segmentation accuracy comparable to full supervision.

**Strengths:**

1. Originality: The use of approximate size targets as a form of weak supervision for semantic segmentation is novel and creative.
2. Quality: The experimental results are comprehensive and demonstrate the effectiveness of the proposed method across different datasets and segmentation architectures.
3. Significance: The approach has significant implications for reducing annotation costs in semantic segmentation tasks, making it highly relevant to practical applications.

**Weaknesses:**

1. Simplicity of Method：While the proposed method is innovative, it seems relatively simple. There might be opportunities to enhance its contributions with further development or by integrating additional techniques.
2. Limited Scope of Evaluation: While the paper evaluates the method on several datasets, it would benefit from a broader range of scenarios, including more diverse and complex images.

**Questions:**

Is it possible to give ablation experiments about these two losses? How much does the two losses affect the performance of the model?

For this method, is it possible to give more information about the time comparisons of the various methods in terms of time spent on labeling?

**Limitations:**

The authors have addressed the limitations related to annotation errors and have demonstrated the robustness of their method to these errors. However, it would be beneficial to discuss potential limitations in more detail, such as the scalability of the method to larger and more diverse datasets, and any assumptions made about the nature of the size target annotations.

---

> ### Author Rebuttal · Authors · 2024-08-05
>
> **Comment 1:** Simplicity of Method: While the proposed method is innovative, it seems relatively simple. There might be opportunities to enhance its contributions with further development or by integrating additional techniques. \
> **Response:** We appreciate the recognition of the simplicity of our method. We disagree with adding additional techniques just to make our work less simple. Simplicity is one of our research philosophies, also known as "Occam's Razor" in ML and many other fields of Science. It is famously encapsulated in the saying by Albert Einstein, "Everything should be made as simple as possible, but not simpler." It is due to simplicity and generality that there are many opportunities to enhance our method and integrate it with other techniques, as suggested by the reviewer. Since our work is the first to propose size targets for semantic segmentation, we focus this paper on fully covering the general properties of our main ideas, which should facilitate their use in later works by us and others.
>
> **Comment 2:** Limited Scope of Evaluation: While the paper evaluates the method on several datasets, it would benefit from a broader range of scenarios, including more diverse and complex images. \
> **Response:** We find that most weakly-supervised semantic segmentation (WSSS) methods use PASCAL and COCO datasets for evaluation. Some prior work only uses one dataset, PASCAL, for evaluation. We acknowledge that some prior work may have been evaluated on other datasets. However, due to resource limitations, we align with the majority of prior WSSS work by using PASCAL and COCO. This facilitates proper comparison with the most relevant prior art in WSSS. Furthermore, to demonstrate the effectiveness of our method, we also tested our approach on a medical dataset and with human annotation. We believe that the comprehensiveness of the presented evaluation is at least on par with related prior works at major conferences. We will include more diverse and complex evaluations in a future journal publication.
>
> **Comment 3:** Is it possible to give ablation experiments about these two losses? How much do the two losses affect the performance of the model? \
> **Response:**  If the reviewer is referring to the size-target loss and the CRF loss in our total loss (Eqn 12), the ablation study can be found in Figure 4 (left), where the red plots indicate the performance with the size-target loss (Eqn 2) and the blue plots present the performance with the total loss (Eqn 12). If the reviewer refers to some other two losses, please clarify which ones so that we can address such concerns.
>
> **Comment 4:** Is it possible to give more information about the time comparisons of the various methods in terms of time spent on labeling? \
> **Response:** The average labeling time comparisons across different forms of weak supervision are illustrated in Figure 1. Note that such information for existing supervision methods is collected from well-known prior papers (cited in our work). For our size supervision, we have provided our average labeling speed in Figure 1. Class-specific speeds are detailed in Table 1 for the cat, dog, and bird classes. We are not sure what else could be useful. If there are some specific ideas, please share them.

---

### Official Review · Reviewer_FTCz · 2024-07-12

**Soundness:** 2
**Presentation:** 2
**Contribution:** 3
**Rating:** 5
**Confidence:** 4

**Summary:**

This paper introduces a novel image-level supervision method for semantic segmentation using approximate segment size targets. It utilizes categorical distributions for expected average predictions, reducing annotation cost and complexity. The authors propose a zero-avoiding KL divergence as a training loss, compatible with any segmentation architecture, and demonstrate significant robustness to size target errors, improving generalization. The method achieves state-of-the-art performance on multiple datasets with standard segmentation models like ResNet101. Additionally, it requires minimal extra information and no architectural changes, making it a practical and effective solution for weakly-supervised semantic segmentation in real-world applications.

**Strengths:**

1. The paper introduces a novel form of image-level supervision for semantic segmentation using approximate segment size targets. This approach is original in its use of categorical distributions for expected average predictions, providing a fresh perspective on weakly-supervised segmentation methods.

2. The quality of the research is high, with comprehensive experiments conducted on multiple datasets. The use of a zero-avoiding variant of KL divergence as a training loss is well-justified and demonstrates robustness to size target errors. The empirical results show that the method achieves state-of-the-art performance using standard segmentation models.

**Weaknesses:**

1. The paper claims robustness to size target errors but provides limited detailed analysis on this aspect. Including more experiments to quantify and analyze how different levels of size target errors impact performance would provide a clearer understanding of the method's robustness.

2. Lack of related work. The paper’s logical flow and organization need improvement.

3. The paper lacks comprehensive comparisons with the latest models, such as "SFC: Shared Feature Calibration in Weakly Supervised Semantic Segmentation (AAAI24)".

**Questions:**

see the Weaknesses

**Limitations:**

1. Fig and Figure are inconsistent in Line 24

---

> ### Author Rebuttal · Authors · 2024-08-05
>
> **Comment 1:** The paper claims robustness to size target errors but provides limited detailed analysis on this aspect. Including more experiments to quantify and analyze how different levels of size target errors impact performance would provide a clearer understanding of the method's robustness. \
> **Response:** We have conducted a detailed analysis of the network's robustness to size errors. The reviewer may have missed these supporting experiments because they are scattered throughout the paper. Here is a summary of all relevant experiments related to robustness. We demonstrate robustness in Figures 4, 5, and 6, showing performance with respect to different levels of size errors (mRE level in our Gaussian noise model). Specifically, Figure 4 shows the robustness of the networks using synthetic sizes with various levels of size errors on Pascal. Figure 5 (right) and Figure 6 (left) demonstrate robustness under similar experimental settings on a subset of Pascal with cat, dog, and bird classes, and on a medical dataset.
>
> **Comment 2:** Lack of related work. The paper’s logical flow and organization need improvement. \
> **Response:**  We have a discussion of related work in the introduction, specifically in Section 1.2. We would be happy to address more specific feedback on issues with the logical flow and organization.
>
> **Comment 3:** The paper lacks comprehensive comparisons with the latest models, such as "SFC: Shared Feature Calibration in Weakly Supervised Semantic Segmentation (AAAI24)". \
> **Response:** Thank you for pointing this out. We will add the SFC results (71.2% on PASCAL and 46.8% on COCO with R101 backbone) to the multi-stage partition in Table 2. However, it's worth noting that our size-target approach is designed to be end-to-end without the necessity of any architectural modifications, similar to fully supervised systems. Due to its simplicity and generality, it is possible to design complex multi-stage methods on top of it. It's not fair to compare our results based on end-to-end standard architectures with multi-stage systems. The multi-stage methods listed in Table 2 are for completeness, but they are not intended as direct performance benchmarks.

---

### Official Review · Reviewer_GY7b · 2024-07-14

**Soundness:** 2
**Presentation:** 2
**Contribution:** 2
**Rating:** 4
**Confidence:** 5

**Summary:**

The paper introduces a novel image-level supervision method for semantic segmentation, utilizing approximate targets for the relative sizes of segments in training images. These targets, represented as categorical distributions for the expected average prediction over pixels, are integrated using a zero-avoiding variant of KL divergence as the training loss. This approach achieves quality comparable to full pixel-level supervision but is significantly less costly, requiring only rough estimates of the areas occupied by each class.

**Strengths:**

1. Using object size as a form of supervision is both innovative and interesting.
2. The proposed method is straightforward and easy to understand.

**Weaknesses:**

1. The title of the paper is misleading. It claims that approximate size targets are sufficient, but the work also uses image labels for supervision.
2. The most important comparison in Figure 1 is between 'Tag' and 'Size target,' as this validates the significance of using target size supervision. To clearly demonstrate that 'Size target' is superior to 'Tag' under identical conditions, it would be better to use the same architecture for both comparisons.
3. Labeling the size of objects can be challenging for humans and may introduce significant noise, especially for tiny objects. Although the authors demonstrate impressive accuracy with up to 8% size target errors, this remains a stringent annotation standard, particularly for small objects. For instance, as seen in Table 1, the mean relative error (mRE) often exceeds 10% during human annotation in the Pascal VOC dataset. Moreover, estimating target sizes in Pascal VOC is relatively easy since objects are typically large and centered. However, labeling images in more complex datasets, such as COCO, might result in a higher mRE.
4. In Table 1, the authors should also report the speed of tag annotation to highlight the cost of estimating target sizes.
5. The proposed method is straightforward and impressive for its end-to-end training, especially considering that existing weakly supervised semantic segmentation (WSSS) methods typically use CAM and two-step training. However, as shown in Table 2, while the proposed method achieves comparable accuracy to state-of-the-art WSSS methods, it relies on additional supervision and a high annotation standard (8% mRE). Moreover, Table 2 indicates that the accuracy with only tag supervision is close to that of fully supervised methods, suggesting that tag supervision alone may be sufficient for segmentation.

**Questions:**

See Q3 and Q5 in weaknesses, pls.

**Limitations:**

The authors do not discuss the limitations and broader impact of their method, which necessitates a dedicated discussion.

---

> ### Author Rebuttal · Authors · 2024-08-05
>
> **Comment 1:** The title of the paper is misleading. It claims that approximate size targets are sufficient, but the work also uses image labels for supervision. \
> **Response:** We would like to reassure the reviewer that there was no intention to mislead the readers about the size-target supervision including class tag information. We tried to be clear about it. For example, the abstract states that size targets are image-level supervision (line 1) **extending** standard class-tag labels (line 11). We believe that the English word "extending" means "enlarging", which implies "inclusion".  We gave an example of the size target v=(0,.15,0,...,0,.75) on line 24 clarifying that it is a categorical distribution over K classes. To further clarify the "inclusion" of tags, we can point out that the corresponding class tags t=(0,1,0,...,0,1) can be easily extracted, e.g. by the "ceiling" operator t = ceil(v). Would that help?
>
> **Comment 2:** Using the same architecture to compare 'Tag' and 'Size target' in Figure 1. \
> **Response:** We understand the reviewer's concern about not using the same architecture for 'Tag' and 'Size target' in Figure 1. Since we already have the 'Size' result on WR38 architecture in Table 2 (72.7%), we will add this number to the "size" column in Figure 1. Then, it can be compared fairly to all other supervisions in this Figure. Thank you for pointing this out.
>
> **Comment 3:** Labeling the size of objects can be challenging for humans and may introduce significant noise, especially for tiny objects. Although the authors demonstrate impressive accuracy with up to 8% size target errors, this remains a stringent annotation standard, particularly for small objects. For instance, as seen in Table 1, the mean relative error (mRE) often exceeds 10% during human annotation in the Pascal VOC dataset. Moreover, estimating target sizes in Pascal VOC is relatively easy since objects are typically large and centered. However, labeling images in more complex datasets, such as COCO, might result in a higher mRE. \
> **Response:** We acknowledge that labeling tiny objects can be challenging. However, this issue is not unique to size supervision and applies to other forms of supervision. We have observed that many ground truth masks in PASCAL are inaccurate for tiny objects (e.g. many void/boundary labels 255). Tiny objects are also very hard to find and identify in tag supervision.
>
> Regarding our choice of 8% MRE for synthetic size being different from 16% MRE for humans... We argue that mRE-matching is wrong. Figure 5 (left) compares our Gaussian noise model using 16% mRE and the human error distribution, also 16% mRE. This reveals that the latter distribution is very different and likely contains "heavy tails", as discussed on line 244. We could have tried to find a better matching "heavy tail" noise distribution, but instead, we chose to simply adjust the mRE of our synthetic Gaussian model to match the performance of the human errors. As shown in Figure 5 (right), 8% mRE Gaussian closely approximates the human error performance (see line 256). We speculate that neural networks are robust to heavy tails, thus larger human mRE statistic (increased by such heavy tails) is largely irrelevant. We thank the reviewer for bringing out these important points. We will gladly clarify them in the paper.
>
> Regarding the comment that "sizing" objects in other datasets like COCO may result in a higher mRE... We think this is debatable as PASCAL is highly diverse and includes objects of many types and shapes: tiny and large, thin and thick, centered and non-centered, simple and complex, single or plural, occluded or not, etc. One clear difference in COCO is a larger number of categories, but our size-annotation tool focuses the user only on one category at a time. It is also possible to further improve our size annotation assistance tool, e.g. by fine-tuning it to objects of each specific class.
>
> **Comment 4:** The authors should also report the speed of tag annotation to highlight the cost of estimating target sizes in Table 1. \
> **Response:**  We will copy the average tag annotation speed from Figure 1 to make Table 1 more self-contained.
>
> **Comment 5:** The method is impressive for its end-to-end training, especially considering that existing WSSS methods typically use CAM and two-step training. However, it relies on additional supervision and a high annotation standard (8% mRE). Moreover, Table 2 suggests that tag supervision alone may be sufficient for segmentation. \
> **Response:**  We appreciate the reviewer's recognition of the strengths of our proposed method, particularly its end-to-end training capability. This question regarding "tag vs. size" is highly relevant and brings up an important point for discussion.
>
> In Table 2, for Pascal the best end-to-end performance with tag supervision (76.2%, using non-standard dual stream architecture) is only marginally weaker than our size supervision (78.1%, using standard backbone), and both are comparable to full supervision (81.4%) on Pascal. However, the best tag-only performance for ene-to-end methods on COCO (51.0%, also dual stream) is considerably lower than 56.3% for our simple size-based approach on a standard backbone, which is much closer to full supervision (60%). This makes it hard to claim that tag-only supervision is sufficiently good for accurate segmentation, at least for the current methodologies. Note that even the best (we could find) multi-stage system for tag-only supervision achieves only 53.7% on COCO. (We recently found the result in the paper "Weakly supervised co-training with swapping assignments for semantic segmentation". It will be added to our Table 2.)
>
> Regarding the "high annotation standard" (8% mRE) mentioned by the reviewer: as detailed in our reply to comment 3, we use 8% mRE for our Gaussian noise as a good match for human error performance. It is not a "higher-than-human" standard.

---

> > ### Comment · Reviewer_GY7b · 2024-08-10
> >
> > The response seems superficial and doesn't address my concerns effectively. Here are the specific issues:
> >
> > In Q1: My issue is that the title is misleading, but the authors primarily discuss the abstract, which doesn't resolve the problem.
> >
> > In Q4: Figure 1 shows the average tag annotation, while Table 1 should reflect the annotation speed for three different classes. How are these related?
> >
> > In Q3: The main question is whether an 8% MRE is challenging for human annotation. It is good for the authors to conduct experiments on three classes, but these classes don't represent all classes in the VOC dataset. Additionally, the mIOU for these three classes differs from the mIOU for the entire dataset, as other classes might influence the predictions for these three. Therefore, using segmentation accuracy for only three classes (Figure 5) might not accurately reflect matching of mREs of human annotation and synthetic data.
> >
> > In Q3: The authors claim that PASCAL is highly diverse, but is there any evidence to support this? COCO is treated as an example, but what about datasets with many images containing multiple instances of the same semantic class, as shown in Figure 1?

---

> ### Author Response · Authors · 2024-08-11
>
> > The response seems superficial and doesn't address my concerns effectively.
>
> Superficial? We tried our best and in good faith, but please consider that in some cases we had to guess what your real concerns are. We hope to gradually reduce misunderstandings.
>
> Q1 is a good example. We had to guess the question presented in the **but** part:
> > “It claims that approximate size targets are sufficient, **but** the work also uses image labels for supervision.”
>
> We interpreted that as a concern that our paper generally hides the fact that size targets are an extension of tags. Our rebuttal addresses this in detail. If you are specifically concerned just about the title, please note that all other forms of weak supervision (boxes, points, etc) also implicitly include tag info, but no prior work explicitly states that in their titles.
>
> We also think that it is rather obvious that size targets “extend” class tags. In any case, the reader does not have to wait too long and this is emphasized right in the abstract. We like our title, but we are also open to your suggestions about it.
>
> > In Q4: Figure 1 shows the average tag annotation, while **Table 1 should reflect the annotation speed for three different classes**. How are these related?
>
> Table 1 has size target annotation speed for three classes individually only because our annotation assistance tool works with one class at a time and we can collect such info separately for each class. All timing info on other forms of weak supervision (in Figure 1) is collected from prior work [46]. We do not have any information about tagging individual classes. This information may not be even possible depending on how the corresponding assistance tools work.
>
>
> > In Q3: The main question is **whether an 8% MRE is challenging for human annotation**.
>
> As we hoped to clarify in the rebuttal, mRE is irrelevant here (it is mainly needed as a scalar tuning parameter controlling our noise model). We believe a more relevant question for our paper is **whether the quality of human size annotation is good enough to train the network to produce the segmentation quality comparable with full pixel-level supervision**. Our experiments conclusively confirm this is possible. The consistency of our results (human and synthetic, across three datasets) is sufficiently convincing, in our opinion.
>
> First, we use (our) human annotation for a subset of PASCAL on cat, dog, and bird classes, individually (see binary segmentation in Table 1) and together (four-class segmentation in Figure 5 right). For example, the latter achieves 89.6% mIOU for all 4 classes, while full supervision is 92.2% for the same four classes. "Bird" is a particularly complex class on PASCAL due to the huge variation in size and plurality of objects. For birds only, human annotation achieves 86.4% accuracy, while full supervision is 88.8%. (Table 1) We are not sure why the reviewer thinks that our classes are not representative of PASCAL
>
> Second, due to limited human resources, we could not size-annotate all PASCAL classes, COCO, or other segmentation datasets. Instead, we developed a synthetic noise model for corrupting true sizes easily available on all these sets. Our noise model was tuned to match the segmentation quality for human annotation on available three PASCAL classes (which vary in complexity). Assuming that our tuning by matching segmentation quality is technically justified (see next point), we provide further evaluations on all PASCAL classes and COCO, which further confirm our claim. Of course, we can not guarantee that our ideas work in 100% of all cases. No one can in computer vision. We could not claim that even if we fully human-annotated PASCAL, COCO, Cityscapes, ADE20K, etc. However, we believe we provided sufficient evidence on real and synthetic experiments for the promising potential of our ideas.
>
> > **Other classes might influence** the predictions for these three (cat, dog, bird).
>
> The following answer is based on a particular guess of the real concern here. When matching the segmentation quality (for synthetic and human annotations) we use the same setting for both (e.g. four-class segmentation). So “other classes” have no influence here. When adding “more classes”, they should make the prediction problem equally more challenging for both human and synthetic annotations. However, “more classes” do not affect the size annotation accuracy in both cases. Indeed, we corrupt target sizes independently for each class, the humans also evaluate only one class at a time. In both cases, we convert such corrupted sizes to a distribution by normalization mainly for the sake of the loss function (KL divergence), but this is the same process for human or synthetic targets.
>
> If this does not help, please elaborate. We will try again.
>
> > prove that images in Pascal are **diverse enough** or **more diverse**
>
> We can not do that. The claim that it is not diverse enough is also speculative. Pls. see the next reply

---

> ### Author Response · Authors · 2024-08-13
> **extra comments on using more "complex" datasets**
>
> We noticed that our response to reviewer thaL may be helpful to clarify our view on the need to test more complex datastes, which is similar to one of your points. However, we realized you might not see our response to that reviewer. Here is a copy of our relevant reply to thaL.
>
> **[for us, complex are classes, not datasets]**:  A motivation for re-evaluating our ideas on more datasets is based on a speculation that human size annotation accuracy will decrease on more "complex datasets", whatever that may mean. Since we ask annotators to size only one class at a time, it makes sense to focus the discussion on some specific class that might be harder to size than classes on PASCAL. Could you please name some particular class on Cityscapes or DE20K that you think is harder to size than “birds” and why you think so? We do not see how image resolution or abstract “scene complexity” is relevant. If “annotation granularity” means high-frequency boundary details, they are mostly irrelevant for size estimation accuracy (due to “averaging”), unless many thin structures dominate the object shape, as often the case with birds.
>
> We agree that harder-to-size classes may exist, but it would be helpful to have a specific example from the datasets you suggested that would make this discussion less speculative. We believe that significantly higher human errors are possible for some extreme examples of classes that this discussion can identify (we can name them in limitations). This may also degrade the corresponding results. However, our results are sufficiently convincing for many representative objects that could be found in practical applications. Things work even for sufficiently challenging classes like birds. One should also keep in mind that better assistance tools could be designed for specific complex classes. In general, our results are meant to be a proof-of-concept and we believe they sufficiently demonstrate the potential of our novel ideas.

---

### Author Rebuttal · Authors · 2024-08-06

We thank the reviewer for their insightful and positive feedback on our size-target approach for image-level semantic segmentation. We are encouraged by their recognition of the novelty (GY7b, FTCz, AEBs, thaL), simplicity (GY7b, AEBs), and practicality (FTCz, AEBs, thaL) of our approach. We are glad many reviewers found our evaluation comprehensive (FTCz, AEBs), the approach clearly presented (GY7b, thaL), and the performance achieved state-of-the-art (FTCz) and comparable to methods using extensive labels (GY7b, AEBs, thaL).

Most of the reviewers' critical comments do not overlap and we address them individually for each reviewer. However, several reviewers indicated their interest in a discussion of the broader impact of our general method and its limitations. Here, this general rebuttal section provides such brief discussion.

Our title is an informal claim that "approximate size targets are sufficient for accurate segmentation". This is not a theorem, yet our main intention is to share with the community a surprising ("impressive", according to GY7b) finding that enriching class tags with approximate size information significantly simplifies the segmentation problem, even though the extra information remains image-level and excludes any object localization. We observe that many standard segmentation architectures can resolve all ambiguities and reach accuracies closely approaching full-mask supervision using only simple losses based on approximate size targets. In contrast, tags-only supervision on PASCAL currently requires complex multi-stage systems to achieve a similarly good quality, while on COCO even complex tag-only systems are significantly worse than what approximate size targets easily get based on standard end-to-end segmentation backbones.

Our new supervision principle for segmentation is general and could be useful in many practical applications - it does not require the design of complex muti-stage systems and avoids prohibitively expensive GT segmentation masks. We found that approximate size annotation is only marginally more complex than tag annotation - it is easy to design assistance tools significantly simplifying size estimation. Moreover, we found that the training process is robust to error levels that one can expect from a human annotator. We also believe that the new supervision principle itself is technically interesting and mathematically elegant. It extends binary tag indicators to soft distributions. It also poses many interesting open questions for further research, e.g. why approximate image-level size information is nearly as useful for neural networks as full pixel-level masks? Or, why is it hard to design simple (e.g. end-to-end) solutions based only on class tags, which seem only slightly weaker than approximate class-size information (particularly compared to pixel masks). In particular, it further stimulates research into better loss functions for tag supervision that can reduce the gap with closely related size supervision. We are also surprised that some noise in size is even beneficial for the quality of training. These findings and questions could be interesting to the community. At least, this is what makes this paper interesting to us.

---

### Decision · Program_Chairs · 2024-09-25

**Decision:**

Reject

**Comment:**

This paper is a borderline case, with reviewers giving scores centered around 5 (more precisely: 4, 5, 5, 6). The reviewers appreciated the idea of using relatively cheap object size supervision in addition to the traditional image-level labels. However, they do not seem convinced that the moderate improvement in performance is large enough to justify the extra manual annotation effort (especially considering that modern image-level supervised methods already reach performance rather close to pixel-level supervised methods).

There is also a closely related work from 2016 proposing a similar idea (i.e. using object size estimates to help weakly supervised learning of object detectors, albeit at the bounding-box level rather than at the segment level): https://arxiv.org/abs/1608.04314. This reduces the novelty of the proposed approach.

When taking all factors into account, the Area Chair recommends rejection.